# Safety and Immunogenicity of the Third Booster Dose with Inactivated, Viral Vector, and mRNA COVID-19 Vaccines in Fully Immunized Healthy Adults with Inactivated Vaccine

**DOI:** 10.3390/vaccines10010086

**Published:** 2022-01-06

**Authors:** Sitthichai Kanokudom, Suvichada Assawakosri, Nungruthai Suntronwong, Chompoonut Auphimai, Pornjarim Nilyanimit, Preeyaporn Vichaiwattana, Thanunrat Thongmee, Ritthideach Yorsaeng, Donchida Srimuan, Thaksaporn Thatsanatorn, Sirapa Klinfueng, Natthinee Sudhinaraset, Nasamon Wanlapakorn, Sittisak Honsawek, Yong Poovorawan

**Affiliations:** 1Center of Excellence in Clinical Virology, Faculty of Medicine, Chulalongkorn University, Bangkok 10330, Thailand; kanokudom_s@yahoo.com (S.K.); suvichada.assawa@gmail.com (S.A.); suntronwong.n@gmail.com (N.S.); chompoonut.bit@gmail.com (C.A.); mim_bhni@hotmail.com (P.N.); preeya_teiy@hotmail.com (P.V.); tata033@hotmail.com (T.T.); ritthideach.yor@gmail.com (R.Y.); donchida.s@gmail.com (D.S.); thaksapohnl@hotmail.com (T.T.); sirapa.klinfueng@gmail.com (S.K.); dr_natthinee@hotmail.com (N.S.); nasamon.w@chula.ac.th (N.W.); 2Osteoarthritis and Musculoskeleton Research Unit, Department of Biochemistry, Faculty of Medicine, Chulalongkorn University, King Chulalongkorn Memorial Hospital, Thai Red Cross Society, Bangkok 10330, Thailand; 3The Royal Society of Thailand (FRS(T)), Sanam Sueapa, Dusit, Bangkok 10330, Thailand

**Keywords:** severe acute respiratory syndrome coronavirus 2 (SARS-CoV-2), booster, third dose, inactivated vaccine, viral vector vaccine, mRNA vaccine, clinical trial

## Abstract

The coronavirus disease 2019 (COVID-19) pandemic has become a severe healthcare problem worldwide since the first outbreak in late December 2019. Currently, the COVID-19 vaccine has been used in many countries, but it is still unable to control the spread of severe acute respiratory syndrome coronavirus 2 (SARS-CoV-2) infection, despite patients receiving full vaccination doses. Therefore, we aimed to appraise the booster effect of the different platforms of vaccines, including inactivated vaccine (BBIBP), viral vector vaccine (AZD122), and mRNA vaccine (BNT162b2), in healthy adults who received the full dose of inactivated vaccine (CoronaVac). The booster dose was safe with no serious adverse events. Moreover, the immunogenicity indicated that the booster dose with viral vector and mRNA vaccine achieved a significant proportion of Ig anti-receptor binding domain (RBD), IgG anti-RBD, and IgA anti-S1 booster response. In contrast, inactivated vaccine achieved a lower booster response than others. Consequently, the neutralization activity of vaccinated serum had a high inhibition of over 90% against SARS-CoV-2 wild-type and their variants (B.1.1.7–alpha, B.1.351–beta, and B.1.617.2–delta). In addition, IgG anti-nucleocapsid was observed only among the group that received the BBIBP booster. Our study found a significant increase in levels of IFN-ɣ secreting T-cell response after the additional viral vector or mRNA booster vaccination. This study showed that administration with either viral vector (AZD1222) or mRNA (BNT162b2) boosters in individuals with a history of two doses of inactivated vaccine (CoronaVac) obtained great immunogenicity with acceptable adverse events.

## 1. Introduction

Severe acute respiratory syndrome coronavirus 2 (SARS-CoV-2) belongs to the family of coronaviruses in the genus of *Betacoronavirus*. SARS-CoV-2 is rapidly transmitted as airborne contagious droplets when in close contact with an infected carrier. Since the beginning of the coronavirus disease 2019 (COVID-19) outbreak in Wuhan, China, in December 2019, this virus had been dramatically transmitted worldwide. The World Health Organization (WHO) declared COVID-19 a pandemic on 11 March 2020. Globally, as of 2 December 2021, over 263 billion confirmed cases of COVID-19 have been recorded, including approximately 5.2 million deaths [1]. Several studies have been reported regarding the efficacy of the COVID-19 vaccine to help minimize disease severity and mortality [2,3,4]. Therefore, the COVID-19 vaccination has been administered on a global scale. Thailand had begun to inoculate most of the population with an inactivated vaccine (CoronaVac; Sinovac Life Sciences, Beijing, China), mainly since the end of February 2021, before the arrival of the viral vector vaccine, AZD1222 (University of Oxford/AstraZeneca, Oxford, UK), from Siam bioscience (AstraZeneca (Thailand) Co., Ltd., Nonthaburi, Thailand). Later on, Thailand launched the first mass vaccination program with many vaccines during June 2021. Furthermore, the inactivated vaccine, BBIBP (Beijing Institute of Biological Products Co., Ltd.; Sinopharm, Beijing, China); viral vector vaccine, Ad26.CoV2.S (Janssen Biotech Inc., Horsham, PA, USA); mRNA vaccine consisting of BNT162b2 (Pfizer-BioNTech Inc., New York, NY, USA); and mRNA-1273 (Moderna, Inc., Cambridge, MA, USA) have been approved by the Thai Food and Drug Administration (FDA) [5].

However, breakthrough infection has been reported despite having received the full dose of CoronaVac in medical healthcare workers [6]. Although the administration of CoronaVac was achieved in Bangkok, we observed that the low vaccine efficacy affected control efforts to limit the spread of SARS-CoV-2 infection [7]. In April 2021, the alpha (B.1.1.7) variant was prominent in Thailand. The delta (B.1.617.2) variant became the most prevalent one after July 2021 [8] and also spread globally, as recognized by WHO and the Centers for Disease Control and Prevention (CDC) [9,10]. Currently, the delta variant has overtaken the original COVID-19 virus strain as the most widespread strain in the pandemic, even in regions with high vaccination rates [10,11,12]. Therefore, the booster dose of SARS-CoV-2 is urgently required to stabilize the rising numbers of infected individuals. The preliminary immunogenicity data from the real-world immunization has indicated that the booster dose of AZD1222 in the CoronaVac cohort elicited a high antibody titer and neutralization activity against SARS-CoV-2 and their variants [13]. In addition, animal model studies have shown that the booster dose with inactivated, adenoviral vector, and mRNA vaccines evoked a high titer of neutralizing antibody [14]. Therefore, in this study, we aimed to investigate the reactogenicity and immunogenicity of the third booster dose with the same and different COVID-19 platform vaccines.

## 2. Materials and Methods

### 2.1. Study Design

Participants were healthy volunteers (age: ≥18 years) who were recruited for vaccination at the clinical trials unit at the Center of Excellence in Virology, Department of Pediatrics, Faculty of Medicine, Chulalongkorn University in Bangkok, Thailand. All participants are Thai citizens who currently live in Bangkok during the clinical trial period. A total of 177 healthy subjects who received two doses of the inactivated COVID-19 vaccine, CoronaVac (CV) (Sinovac Biotech Co., Ltd., Beijing, China), were divided into three groups. The first group received the inactivated vaccine BBIBP-CorV (Beijing Institute of Biological Products Co., Ltd.; Sinopharm, Beijing, China) (n = 60); the second group received the viral vector vaccine ChAdOx1-S/nCoV-19, also called AZD1222 (University of Oxford/AstraZeneca, Oxford, UK) (n = 57); and the third group received an mRNA vaccine called BNT162b2 (Pfizer-BioNTech Inc., New York, NY, USA) (n = 60) at 3–4 months after receipt of the first dose (Appendix A). After receiving the additional dose, all study participants were monitored for safety and immunogenicity.

The study protocol was approved by the Institutional Review Board (IRB), Faculty of Medicine, Chulalongkorn University (IRB number 546/64), and this trial was registered with the Thai Clinical Trials Registry (TCTR 20210910002). Informed consent was obtained before participant enrollment. The study was conducted according to the Declaration of Helsinki and the principle of Good Clinical Practice Guidelines (ICH-GCP).

The participants were initially recruited after ensuring that there was no history of SARS-CoV-2 infection. Blood samples were collected prior to vaccination (day 0, baseline) and after receiving the booster dose (day 14 and day 28).

### 2.2. Vaccines

BBIBP-CorV (referred to as BBIBP) is an inactivated vaccine developed from whole SARS-CoV-2 stain HB02. Briefly, the HB02 strain is obtained by passaging and purification in Vero cells. Consequently, the whole virion is inactivated in β-propionolactone and further absorbed with aluminum hydroxide. One dose (0.5 mL) contains 6.50 U [15].

ChAdOx1-S/nCoV-19 (referred to as AZD1222) is a non-replicating chimpanzee adenovirus Oxford 1 vector vaccine presenting the SARS-CoV-2 spike protein (AZD1222). The virion is produced in genetically modified HEK293 cells. One dose (0.5 mL) contains 5 × 1010 infectious units [16].

BNT162b is a lipid nanoparticle containing modified RNA encoding the SARS-CoV-2 full-length spike, modified by two proline mutations to lock it in the prefusion conformation. One dose (0.3 mL) contains 30 μg [17].

### 2.3. Reactogenicity Assessment

The subject was continuously monitoring for details of AEs following immunization (AEFIs) within 7 days by self-assessment records via online or paper-based questionnaires. An explanation about data collection was given to participants by trained investigators during the initial visit, and local, systemic, and any AEs were recorded.

### 2.4. Laboratory Measurements

#### 2.4.1. Immunoglobulin and IgG Anti-RBD Assays

The serum fraction from the participant was analyzed for total immunoglobulin (Ig) specific to the receptor-binding domain (RBD) of the SARS-CoV-2 spike (S) protein using an Elecsys SARS-CoV-2 S electrochemiluminescence immunoassay (ECLIA) according to the manufacturer’s instructions (Roche Diagnostics, Basel, Switzerland). The quantitative immunoglobulin IgG titer was reported as units per milliliter (U/mL). The IgG anti-RBD was tested using a SARS-CoV-2 IgG II Quant chemiluminescent microparticle immunoassay (CMIA) (Abbott Laboratories, Abbott Park, IL, USA) according to the manufacturer’s instructions. The results are quantitative and given as arbitrary units per milliliter (AU/mL). Then, the value was multiplied by 0.142 to convert it to binding antibody units per milliliter (BAU/mL).

#### 2.4.2. IgG Anti-N Assay

For the SARS-CoV-2 anti-nucleocapsid (anti-N) IgG, the serum fraction was also tested using the CMIA (Abbott Diagnostics, Sligo, Ireland). The semi-quantitative results are reported in the unit of sample per calibrator or index (S/C) and were interpreted following the manufacturer’s instruction. For interpretation, S/C ≥ 1.4 was defined as positive and S/C < 1.4 as negative.

#### 2.4.3. IgA Anti S1 Assay

The SARS-CoV2-2 anti-S1 IgA was monitored using an enzyme-linked immunosorbent assay (ELISA) (Euroimmun, Lübeck, Germany). Each kit contained microplate strips with 8 break-off reagent wells coated with recombinant structural proteins of SARS-CoV-2. Briefly, in the first reaction step, diluted patient samples are incubated in the wells. For positive samples, specific antibodies will bind to the antigens. A second incubation is carried out using an enzyme-conjugated antihuman IgA catalyzing a color reaction to detect the bound antibodies. The color intensity after the stop reaction was evaluated at 450 nm. The semi-quantitative analysis can be interpreted by calculating a ratio of optical density between the sample and the calibrator. The upper limit ratio (S/C) > 9 was reported as 9.0.

#### 2.4.4. Neutralization Assay

Serum samples were also evaluated for neutralizing activity against the SARS-CoV-2 wild-type and variants of concern (VOCs), namely B.1.1.7 (alpha), B.1.351 (beta), and B.1.617.2 (delta), using an ELISA-based surrogate virus neutralization test (sVNT). Additionally, a cPass SARS-CoV-2 neutralizing antibody detection kit (GenScript, Piscataway, NJ, USA) was used for all strains. The recombinant RBDs from B.1.1.7 (containing N501Y), B.1.351 (containing N501Y, E484K, and K417N), and B.1.617.2 (containing L452R and T478K) were also used with this kit. Briefly, the serum samples were diluted 1:10 with buffer and incubated with RBD conjugated to horseradish peroxidase for 30 min at 37 °C. Next, 100 µL of the sample mixture was added to a capture plate pre-coated with human angiotensin-converting enzyme 2 (ACE2) and incubated for 15 min at 37 °C. After washing, 100 µL of 3,3′,5,5′-tetramethylbenzidine (TMB) solution was added and the plate was incubated in the dark for 15 min at room temperature. After the addition of 50 µL stop solution, samples were read at 450 nm. The ability of a serum to inhibit binding between RBD and ACE2 was calculated as a percentage as follows: 1—(average OD of sample/average OD of negative control) × 100.

#### 2.4.5. SARS-CoV-2 Stimulating IFN-γ Assay

Heparinized whole blood was collected in a blood collection tube following the manufacturer’s instruction and incubated at 37 °C for 21 h (QuantiFERON Human IFN-γ SARS-CoV-2, Qiagen). This assay consisted of two sets of SARS-CoV-2 S antigens. The Ag1 tube contained CD4+ epitopes derived from the S1 subunit (RBD), while the Ag2 tube contained CD4+ and CD8+ epitopes from the S1 and S2 [18]. Moreover, the plasma fraction was used to determine the concentration of IFN-γ (IU/mL) using QuantiFERON^®^ ELISA. Absorbance was measured at 450 nm and calculated using IFN-γ standard curve using the QuantiFERON R&D Analysis Software. The QFN ELISA’s detection limit was 0.065 IU/mL, while the maximum limit was 10.0 IU/mL.

### 2.5. Statistical Analysis

The sample size was calculated using G*power software version 3.1.9.6 (based on conventional effect size = 0.25, given significance level (α) = 0.05, power (1-β) = 0.8, numerator degree of freedom = 2, and number of groups = 3). The graphical representation and statistical analyses were carried out using GraphPad Prism version 7.0 for Microsoft Windows. Categorical analyses of age and sex were performed using the chi-square test and Welch’s ANOVA. IgG-specific RBDs were designated as geometric mean titers (GMT) with a 95% confidence interval (CI). Other parameters were presented as medians with interquartile ranges. The differences in antibody titers, S/C, and percentage inhibition and IU/mL minus nil between groups were calculated using the Kruskal–Wallis test or Wilcoxon signed-rank test (non-parametric) with multiple comparison adjustments. A *p*-value < 0.05 was considered to indicate statistical significance.

## 3. Results

### 3.1. Demographic Data

Healthy participants aged ≥18 years who received two doses of CV were recruited by physicians or trained research nurses for additional doses with inactivated, viral vector, or mRNA COVID-19 vaccine after 3–4 months from the first dose of CV. All of the 177 enrolled participants had no or well-controlled underlying diseases, were not in the immunocompromised status, or had received any ongoing immunosuppressive therapy. The baseline demographic characteristics of all participants in the three groups, namely those who received BBIBP (n = 60), AZD1222 (n = 57), or BNT162b2 (n = 60) vaccines, were generally comparable (Table 1). The CV vaccine was originally administered as a two-dose regimen at 21–28 days apart to the Thai individuals aged between 18 and 59 years. The average age of the enrolled participants for the booster dose was 42.9 (IQR: 36–49) years. The mean ages of participants who received BBIBP, AZD1222, and BNT162b2 were 42.7, 41.6, and 44.2 years, respectively. No significant differences in age and sex were observed among the study groups. The underlying diseases of the enrolled participant are reported in Table 1. In the AZD1222 regimen, one participant was excluded as they developed SARS-CoV-2 infection, which was validated by nasopharyngeal swab and real-time polymerase chain reaction on day 7 after vaccination. This participant was asymptomatic until leaving the alternative quarantine station. Additionally, one patient was lost to follow-up during the study period, while another one who received the BNT162b2 vaccine was also lost to follow-up (Table 1).

### 3.2. Reactogenicity Data for the Cohort Receiving a Different Type of SARS-CoV-2 Vaccine

Between March 2021 and May 2021, the Thai participants who received two doses of CV approximately 3–4 months apart were recruited for a third booster vaccine. The reactogenicity data for this cohort were significantly lower with respect to solicited local and systematic adverse events (AEs) in those who received with inactivated vaccine (BBIBP) as compared with others. Moreover, most participants who received BBIBP developed minimal post-vaccination AEs (Figure 1A). The most common solicited local AE after the booster was pain at the injection site: BBIBP group (36.7% mild, 5.0% moderate AE had peaked on day 0) (Figure 1A); AZD1222 group (52.6% mild, 26.3% moderate, and 5.3% severe AE had peaked on day 1) (Figure 1B); BNT162b2 (60.0% mild, 30.0% moderate, and 1.7% severe AE had peaked on day 1) (Figure 1C). The most common systemic AE after the booster dose was myalgia: BBIBP group (21.7% mild, 3.3% moderate AE had peaked on day 0) (Figure 1A); AZD1222 group (43.9% mild, 22.8% moderate, 5.3% severe AE had peaked on day 1) (Figure 1B); BNT162b2 group (38.3% mild, 16.7% moderate; 1.7% severe AE had peaked on day 1) (Figure 1C). The local and systemic AEs were reduced within a few days after vaccination (Figure 1). Further analysis revealed that the AEs after taking the BBIBP vaccine were considerably lower than the other two vaccine types, except for vomiting. Joint pain was substantial in 40.4% of the total participants receiving the AZD1222 vaccine (Appendix A).

### 3.3. Antibody Assay after Booster Dose with a Different Type of SARS-CoV-2 Vaccine

To investigate the total immunoglobulin anti-RBD of SARS-CoV2, mainly IgG and IgG anti RBD were compared among all groups using the Kruskal–Wallis test with multiple comparison adjustment (Figure 2A,B, Appendix A). The geometric mean titer (GMT) of the total Ig anti-RBD and IgG anti-RBD values of the three groups showed similar levels before the booster dose (baseline, day 0). The third dose with BBIBP, AZD1222, or BNT162b2 vaccines significantly elicited total Ig anti-RBD GMTs for 1073, 9865, or 20,787 U/mL at 14 days post vaccination, respectively (*p* < 0.0001). Moreover, the immunoglobulin anti-RBD was slightly reduced at 28 days to determine the level of IgG anti-RBD. The trend of IgG anti-RBD was consistent with the immunoglobulin anti-RBD results (Figure 2A,B, Appendix A). The data indicated that the BNT162 vaccine gave the highest level of immunization, whereas BBIBP gave the lowest level of immunization. Compared to the antibody titer at 14 days between the AZD1222 and BNT162b recipients, we found that there were no significant differences in the GMTs of the total Ig anti-RBD and IgG anti-RBD.

The median result showed that IgG anti-N (nucleocapsid) was seronegative (cut-off < 1.4) at baseline. The data showed that a total of 177 participants were not previously infected with SARS-CoV-2 before the booster dose. Moreover, only BBIBP could significantly elicit IgG anti-N (Figure 2C, Appendix A). Additionally, the median of IgG anti-N could be detected, although below the cut-off ratio in the AZD1222 and BNT162b groups. The residual IgG anti-N was possibly obtained from the previous two doses of CoronaVac (Figure 2C).

There were no significant differences in IgA anti-S1 among the three patient groups at baseline. Interestingly, the booster dose could significantly increase IgA anti-S1 (*p* < 0.0001) after receiving AZD1222 and BNT162b. Although BBIBP could elicit over 2-fold increases in the IgA anti-S1 ratio at 14 days compared with baseline, the difference was not significant (Figure 2D, Appendix A). Additionally, the IgA anti-S1 level decayed at day 28 with Ig/IgG anti-RBD (Figure 2, Appendix A).

### 3.4. Neutralization Assay against SARS-CoV-2 Wild-Type and Variants of Concern

A surrogate virus neutralization assay was performed to determine whether the vaccinated serum had functionally inhibited the SARS-CoV-2 binding. The data regarding neutralization activity against SARS-CoV-2 of vaccinated serum were in accordance with the Ig RBD and IgG RBD results. The results showed that the booster with either AZD1222 or BNT162b2 had inhibition activity against wide-type (WT) and variant strains at over 90% on day 28, whereas the booster with BBIBP was significantly lower in inhibition activity against SARS-CoV-2 WT and their variant strains than the two other two vaccines (Figure 3A–C, Appendix A). There was no comparison of neutralizing activity levels between the AZD1222 and BNT162b groups, as the neutralization inhibition was presented at the upper limit of the assay.

According to the data, the booster with either viral vector or mRNA was more favorable given the high immunogenicity against SARS-CoV-2 WT and their variants.

### 3.5. SARS-CoV-2 IFN-ɣ Stimulation

To address whether the vaccination generated a T-cell response, a whole-blood interferon–gamma release assay (IGRA) using QuantiFERON (QFN) SARS-CoV-2-ELISA assay was performed. Multiple comparisons using Kruskal–Wallis’s test for IFN-ɣ CD4+ and IFN-ɣ CD4+ CD8+ showed that there were significant increases in vaccination cohort with AZD1222 (*p* < 0.0001) and BNT162b2 (*p* < 0.0001) within 14 days as compared to baseline, except that BBIBP was not significantly different. Overall, a T-cell response against SARS-CoV-2 peptides was pronounced in the AZD1222 and BNT162b2 cohort (Figure 4A,B, Appendix A).

## 4. Discussion

Given the ongoing threat of the SARS-CoV-2 pandemic and its global transmission, vaccines for COVID-19 were urgently developed in early 2021 and authorized for administration to the eligible population worldwide. The driven heterologous COVID-19 vaccines are currently considered in terms of safe and efficiency. This prospective study aimed to evaluate whether the combination of COVID-19 vaccines are safe and required for long-term protection against SARS-CoV-2. The additional booster dose was allowed to be given to the healthy adult population in multiple countries [20,21]. Our data confirmed that mixed and matched vaccines with a complete primary dose of CV followed by BBIBP, AZD1222, or BNT162b2 were safe and without serious AEs.

Several prior research studies have focused on the antibody responses elicited by the third dose of a similar booster vaccine [22,23,24]. Recent data have shown that a complete dose of CV could not efficiently activate an immune response, manifested by low antibody (GMT approximately 100 U/mL at 4 weeks after vaccination) [13,25,26] and neutralizing activity (approximately 70% WT and nearly 50% variants of concern (VOCs)) levels [13]. Consistent with our study, the baseline characteristics of their participants after 2–3 months showed lower Ig/IgG anti-RBD and neutralizing activity levels against SARS-CoV-2 and their variants. Indeed, the third dose with any platform of the COVID-19 vaccine ensured higher immunogenicity. Our cohort study of AZD1222 booster was also similar to that previously reported in a real-world study [13]. Additionally, the antibody response after a booster with intramuscular BNT162b2 in our current study (3821 BAU/mL) was similar to previous studies (3884 BAU/mL) [27]. Previous studies have also shown that the mix and match of the platform, namely CV/AZD1222, significantly elicited IgG anti-RBD at 28 days compared to CV/CV [28]. Furthermore, AZD1222/BNT162b synergistically promoted IgG anti-spike at both 28 days [29] and 73 days [30]. In addition, the booster vaccine, especially with AZD1222 and BNT162b2, could develop an immune response both IgA anti-spike and IFN-ɣ stimulation. In accordance with this study, the previous report revealed that the group booster BNT162b enhanced immunoglobulin subclasses IgG and IgA and significantly stimulated IFN-ɣ [30]. Intriguingly, sera IgA anti-spike became dominant in individuals vaccinated with the booster. Previous reports have shown that serum IgA acted as a potent and early SARS-CoV-2–neutralizing agent [31,32]. Nevertheless, we did not observe the titer of vaccine-induced mucosal IgA. The reason for this is supported by the evidence that IgA presents faster than IgG at the mucosal site, and most of the tissue would play an important role as a first-line barrier against infection [32,33]. However, it remains unclear whether AZD1222 and BNT162b2 vaccines induced serum IgA [30,34].

Our study has shown that two doses of inactivate vaccines comprising CV/CV and followed by an additional booster with BBIBP at 3 month intervals can significantly elicit the IgG anti-N, along with as SARS-CoV-2 natural infection. The result can be explained by the fact that the inactivated vaccines CV and BBIBP have the SARS-CoV-2 nucleocapsid, while immunization with either AZD1222 or BNT162b2, which do not contain nucleoprotein, did not exhibit any IgG anti-N response. The coronavirus nucleoprotein has been reported to be involved in various aspects of virus replication. A confocal microscopy analysis showed that the avian infectious bronchitis virus N protein localizes both to the cytoplasmic and nucleolar compartments [35]. Rationally, it has been documented that IgG anti-N is functionally involved in humoral immune response [36], interfering with nucleoprotein localization [34]. Moreover, additional evidence has shown that anti-N sera act in intracellular neutralization required by the cytosolic Fc receptor and E3 ubiquitin ligase TRIM21 [37].

The extracellular neutralization of SARS-CoV-2 and their variants (B.1.1.7, B.1.351, B.1.617.2) was also validated by sVNT. This recent study outlined that vaccinated sera could neutralize WT > B.1.1.7 > B.1.617.2 > B.1.351 in a similar pattern with three boosters of inactivated vaccine [38], consistent with previous findings in a real-world study [13].

IFN-ɣ is a potential immunoregulatory protein that facilitates viral clearance brought about by T cells, including T helper type 1 (Th-1) cells, cytotoxic T lymphocytes (CTLs), and natural killer (NK) cells. The current study has shown that T cells produce IFN-ɣ early, by 14 days, which was notably observed in both the BNT162b and AZD1222 groups, although not the BBIBP group. Previous studies showed that an AZD1222 vaccination generated CD4+- and CD8+-mediated IFN-ɣ production within 14 days [39]. BNT162b2 vaccination resulted in the detection of IFN-ɣ CD4+ and CD8+ cells at 29 days post-booster using the ELISpot assay [40]. Furthermore, evidence has shown that there were supportive T-cell responses in the heterologous AZD1222/BNT162b2 participants [29]. In the present study, the third booster of inactivated BBIBP vaccine stimulated mild CD4+ and CD8+ induction of IFN-ɣ when compared with the two others described above.

The recent evidence has shown that the neutralizing antibodies gradually decrease after two doses of inactivated vaccine, then significantly increase after receiving a booster dose. A previous study reported that the memory of IFN-γ T-cells against S, N, M, and O antigens of SARS-CoV-2 can rapidly awaken after receiving a third dose of inactivated vaccine [38]. The latest VOC is known as omicron variant B.1.1.529. This variant is characterized by over 30 changes in spike protein, in particularly 15 changes in RBD, resulting in an evasion of host cell recognition and targeted immune response. Many of the changes may potentially increase the transmissibility of the omicron variant compared to the delta and alpha variants [41]. Additionally, sera from two-dose vaccinated individuals were less capable of neutralizing the omicron variants compared to the prototype and other VOCs [42,43]. Further studies addressing the effectiveness of COVID-19 vaccine against the omicron variant is warranted. The advantages of the inactivated vaccine in countering the new variant are still under investigation.

A number of caveats need to be mentioned in this study. First, the relatively small sample size limits the statistical power of our results. Therefore, multi-center prospective longitudinal investigations with larger sample sizes should be undertaken to determine whether the data show more reliability and whether rare adverse events may be observed, such as thrombosis in AZD1222 and myocarditis in BNT162b2. Second, the combinations and primer–booster intervals of enrolled participants varied in individual studies. Third, we did not use live virus focus reduction neutralization tests (FRNTs) in these subjects. However, the neutralizing activity against SARS-CoV-2 WT and their variant strains was examined using an ELISA-based surrogate virus neutralization test. Additionally, there were limitations in the quantification of serum IgA values, as some of the values exceeded the upper limit of quantification. Lastly, we did not evaluate the mucosal IgA that represent the first line barrier against SARS-CoV-2 internalization. Further studies will be needed to overcome these limitations.

## 5. Conclusions

The clinical trial of the third booster was a practical design used to propose guidelines and build confidence in over 40 countries that administer vast quantities of CoronaVac vaccines. The combination with different platforms as in the present study has yielded immune response against the COVID-19 outbreak with acceptable AEs. Last, the safety and efficacy of heterologous vaccine regimens can provide guidelines for vaccination practice.

## Figures and Tables

**Figure 1 vaccines-10-00086-f001:**
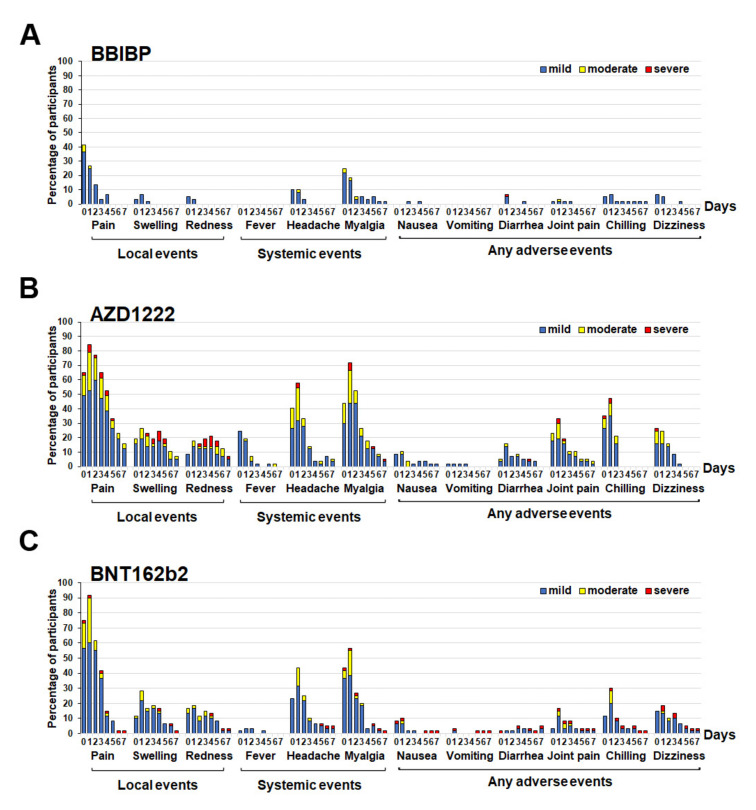
Reactogenicity of a booster dose of SARS-CoV-2 vaccines within 7 days of vaccination. A booster dose of the inactivated vaccine BBIBP (**A**), viral vector vaccine AZD1222 (**B**), and mRNA vaccine BNT162b2 (**C**). The percentages of participants who recorded local, systemic, and any adverse events are shown on the Y-axis. Fever was defined as mild: 38.0 °C to <38.5 °C; moderate: 38.5 °C to <39.0 °C; severe: ≥39.0 °C. For local and systemic symptoms, grading was classified as mild—easily tolerated with no limitation on regular activity; moderate—some limitation of daily activity; severe—unable to perform regular daily activity [19].

**Figure 2 vaccines-10-00086-f002:**
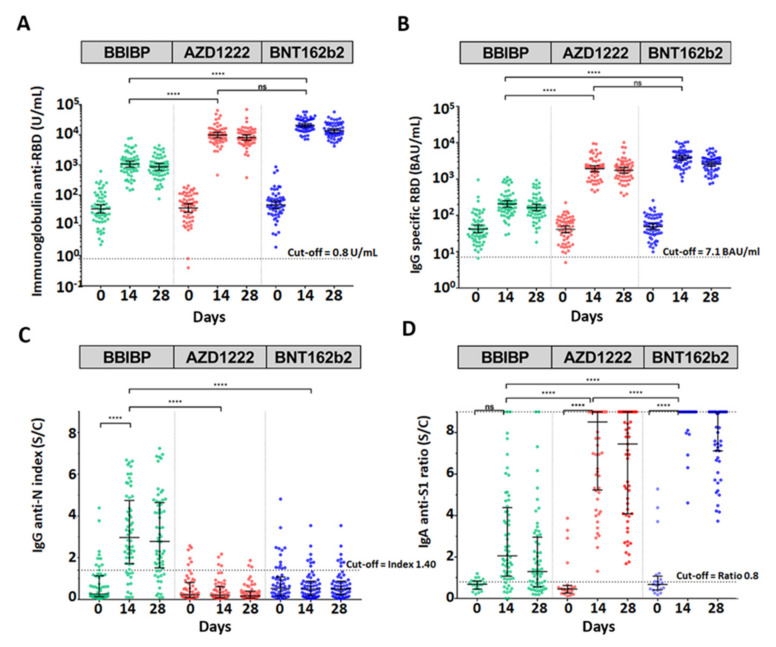
Antibody responses against SARS-CoV-2 assay. The circulating total immunoglobulin anti-RBD of SARS-CoV-2 (U/mL) (**A**). The circulating IgG-specific RBD of SARS-CoV-2 (BAU/mL) (**B**). IgG anti-N of SARS-CoV-2 index (S/C) (**C**). IgA anti-S1 of SARS-CoV-2 ratio (S/C) (**D**). The serum samples were obtained from participants who received two completed doses of the inactivated vaccine, CoronaVac; followed by the inactivated vaccine, BBIBP (green); the viral vector vaccine, AZD1222 (red); or the mRNA vaccine, BNT162b2 (blue), at 3–4 months after the first dose. Lines represent GMTs (95% CI); ns indicates no statistical difference; *p* < 0.0001 (****).

**Figure 3 vaccines-10-00086-f003:**
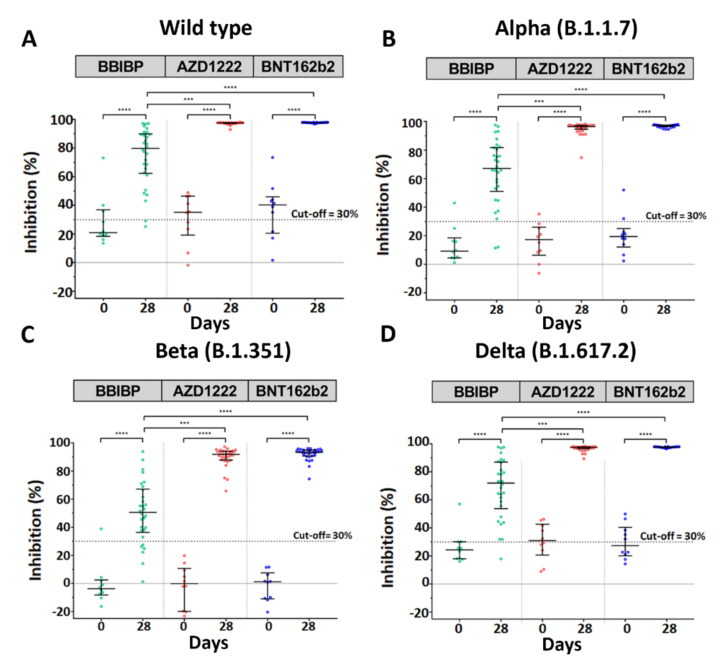
Neutralization activities against wild-type and SARS-CoV-2 variants measured by surrogate virus neutralization test (sVNT). The serum samples obtained from participants who received two completed doses of the inactivated vaccine, CoronaVac; followed by the inactivated vaccine, BBIBP (green); the viral vector vaccine, AZD1222 (red); or the mRNA vaccine, BNT162b2 (blue), at 3–4 months after the first dose were compared. The neutralizing activities against SARS-CoV-2 Wide-type (**A**), Alpha (B.1.1.7) (**B**), Beta (B.1.351) (**C**), and Delta (B.1.617.2) (**D**) were shown. Lines represent medians with interquartile ranges (IQR); ns indicates no significant difference; *p*-value < 0.001 (***), 0.0001 (****).

**Figure 4 vaccines-10-00086-f004:**
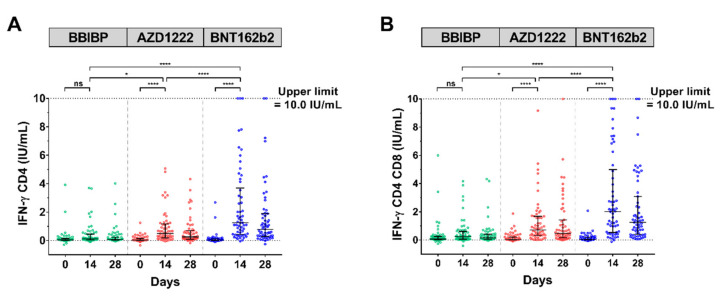
SARS-CoV-2-stimulating IFN-ɣ assay. The heparinized samples were obtained from participants who received two completed doses of the inactivated vaccine, CoronaVac; followed by the inactivated vaccine, BBIBP (green); the viral vector vaccine, AZD1222 (red); or the mRNA vaccine, BNT162b2 (blue), at 3–4 months after the first dose and incubated in a QFN blood collection tube for 21 h. The plasma fraction was evaluated by QFN IFN-ɣ ELISA. The IFN-ɣ produced by CD4-specific Ag1 (**A**). The IFN-ɣ produced by CD4- and CD8-specific Ag2 (**B**). Lines represent medians (IQR); ns indicates no significant difference; *p* < 0.05 (*), 0.0001 (****).

**Table 1 vaccines-10-00086-t001:** Demographics and characteristics of the vaccinated cohorts.

	BBIBP	AZD1222	BNT162b2
**Total number (n) of participant**	60	57	60
**Mean age (year, range)**	42.7 (20–62)	41.6 (21–59)	44.2 (25–58)
**Sex**			
**Male (%)**	30/60 (50.0%)	29/57 (50.9%)	24/60 (40.0%)
**Female (%)**	30/60 (50.0%)	28/57 (49.1%)	36/60 (60.0%)
**Underlying disease (%)**			
**Allergy**	5/60 (10.0%)	2/57 (3.5%)	4/60 (6.7%)
**Breast cancer #**	–	–	1/60 (1.7%)
**Cardiovascular diseases**	1/60 (1.7%) ^1^	2/57 (3.5%) ^2^	–
**Diabetes Mellitus**	1/60 (1.7%)	2/57 (3.5%)	–
**Dyslipidemia**	4/60 (6.7%)	1/57 (1.8%)	6/60 (10%)
**Hypertension**	4/60 (6.7%)	2/57 (3.5%)	2/60 (3.3%)
**Other (Gastritis, Migraine, Thyroid disease, etc.) #**	2/60 (3.3%)	3/57 (5.2%)	2/60 (3.3%)
**Follow-up**			
**Second visit (two weeks)** **Mean (day, range)** **SARS-CoV-2 infection (n)** **Lost to follow-up (n)**	14.2 (13–18) 0 0	14.1 (14–17) 1 0	14.1 (14–19) 1 0
**Second visit (two weeks)** **Mean (day, range)** **SARS-CoV-2 infection (n)** **Lost to follow-up (n)**	28.3 (25–31) 0 0	27.9 (27–28) 1 1	28.1 (28–31) 0 1

# Inactive disease—no medication involving immunosuppressant. ^1^ coronary artery disease; ^2^ heart disease.

## Data Availability

The datasets generated and analyzed during the current study are available from the corresponding author on reasonable request.

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
