# Peer review of "Safety and Immunogenicity of the Third Booster Dose with Inactivated, Viral Vector, and mRNA COVID-19 Vaccines in Fully Immunized Healthy Adults with Inactivated Vaccine"

_vaccines, 2022, doi:10.3390/vaccines10010086_

Round 1

Reviewer 1 Report

Justifications for sample size calculations should be reported in details.

Characteristics of the studied population should be described on details.

According to the realworld data, sample size seems to be so small.

Author Response

1.Justifications for sample size calculations should be reported in details.

Response 1: The sample size is calculated using the G*Power software version 3.1.9.6 (Kang H. J Educ Eval Health Prof. 2021;18:17. doi:10.3352/jeehp.2021.18.17)

Due to the clinical trial composed with 3 group, we added the value for sample size calculation as follows,

F tests - ANOVA: Fixed effects, special, main effects and interactions

Analysis:     A priori: Compute required sample size 

Input:                         Effect size f                         =      0.25

                                   α err prob                            =      0.05

                                   Power (1-β err prob)           =            0.8

                                   Numerator df                      =            2

                                   Number of groups               =            3

Output:                      Noncentrality parameter λ  =      9.8750000

                                   Critical F                             =      3.0543850

                                   Denominator df                   =            155

                                   Total sample size               =            158

                                   Actual power                      =      0.8021998

According to the result of G*power program, the calculated total sample was 158 so we design to enroll as approximately 180 participants (n=60/group) with cover 10% drop-off.

2.Characteristics of the studied population should be described on details.

Response 2: We have already mentioned in line 191-202 and statistic calculation for the age and sex has been analyzed by the Chi-square test and Welch’s ANOVA.

3.According to the realworld data, sample size seems to be so small.

Response 3: This present study conducted a clinical trial based on convenient sampling which composed with three booster groups of different vaccines including the first group received the inactivated vaccine BBIBP-CorV (n = 60), the second group received the viral vector vaccine AZD1222 (n=57), and the third group received the mRNA-BNT162b2 (n=60).

Reviewer 2 Report

The manuscript submitted by Kanokudom et al. entitled "Safety and immunogenicity of the third booster dose with inactivated, viral vector, and mRNA COVID-19 vaccines in fully immunized healthy adults with inactivated vaccine" aims to show that administration with either viral vector (AZD1222) or mRNA (BNT162b2) boosters in individuals with a history of two doses of inactivated vaccine (CoronaVac) obtained great immunogenicity with acceptable adverse events. In this cohort study, three groups of healthy Thai adult were enrolled (age: ≥18 years), with a total of 177 healthy volunteers, who received two doses of the inactivated COVID-19 vaccine, CoronaVac (CV) (Sinovac Biotech co., LTD., Beijing, China), 87 were divided into three groups. The first group received the inactivated vaccine BBIBP-CorV (Beijing Institute of Biological Products Co., Ltd.; Sinopharm, Beijing, China) (n=60) the second group received the viral-vector vaccine– ChAdOx1-S/nCoV-19, also called AZD1222 (University of Oxford/AstraZeneca, Oxford, UK) (n=57); and the third group received an mRNA vaccine called BNT162b2 (Pfizer-BioNTech Inc, NY) (n=60).

The Originality of the study is high, as well as, the significance and the quality of the presented results. Indeed, in nowadays, studies that may contribute to better understand the biology of COVID-19, as the immuno responses to vaccines have great merit.

The results are presented in a clear way and also very well discussed. However, the authors should perform the following improvements before the acceptance of their work:

1 - In the Introduction, the authors should present the new variant omicron;

2 - In M&M section, the authors should justify why the  volunteers received the additional dose after 3-4 months. For example, in Europe, the people are receiving the boost 6 months later, as recomended by OMS and vaccine suppliers

3 - The authors should better emphasized in the text that the average age of the enrolled population around 41-44 years and discuss this fact

4 - If possible, authors should measure early biomarkers of kidney/heart damage after vaccine boosts, to check if vaccines can cause lesions in specific organs

5 - It will be fantastic, if the authors can add same data on the IgA production before and after immunization, since the mucosal IgA is the first barrier against SARS-CoV-2 internalization.

Author Response

1 - In the Introduction, the authors should present the new variant omicron;

Response 1: we appreciated with your suggestion but in our study period the omicron variant have not been yet outbreak, so we did not perform the serological testing on this variant. In our opinion, it not necessary to address the information of this variant in the introduction section. We added a few sentences in discussion that further study on the effectiveness of COVID-19 vaccine against omicron variant is warranted in line 388-390.

2 - In M&M section, the authors should justify why the volunteers received the additional dose after 3-4 months. For example, in Europe, the people are receiving the boost 6 months later, as recomended by OMS and vaccine suppliers

Response 2: From the rational and preliminary studies, we have shown that breakthrough infection has been reported despite having received the full dose of CoronaVac in medical healthcare workers SARS-CoV-2 (as mentioned in line no 68-69). Moreover, mean neutralizing rates at 10–12 weeks post vaccination were 48.0% (95% CI 39.9%–56.1%) against the wild-type strain, 21.8% (95% CI 37.8%–43.9%) against Alpha variant, 1.2% (95% CI 3.5%–8.8%) against Beta, and 1.0% (95% CI 2.9%–7.5%) against Delta (Hunsawong, T et al, Emerging Infectious Disease journal 2021; doi: 10.3201/eid2712.211772). The European are mostly vaccinated with viral vector or mRNA vaccines that it has higher effectiveness compared to CoronaVac (Tregoning JS, et al. Nat Rev Immunol. 2021 doi: 10.1038/s41577-021-00592-1.

3 - The authors should better emphasized in the text that the average age of the enrolled population around 41-44 years and discuss this fact

Response 3:  We added a few sentences in line no. 195-198.

4 - If possible, authors should measure early biomarkers of kidney/heart damage after vaccine boosts, to check if vaccines can cause lesions in specific organs

Response 4: The suggestion is useful. However, the measurement of early biomarkers is not stated in protocol.

5 - It will be fantastic, if the authors can add same data on the IgA production before and after immunization, since the mucosal IgA is the first barrier against SARS-CoV-2 internalization.

Response 5: We did not collect the specimen for mucosal IgA detection. We addressed in the limitation of the study in the discussion section.

Round 2

Reviewer 1 Report

Sample size calculation justification shall be added in one sentence in the method part.

Studied population, the population sample recruitment has been performed from, shall be described. e.g. patients referred to a specific hospital for a specific perpous, volunteer health care providers, workers in an institution.

Author Response

Response (Rounds 2) to Reviewer 1 Comments
Comments and Suggestions for Authors
1.Sample size calculation justification shall be added in one sentence in the method part.

Response 1: the sample size calculation justification has been added in line no. 181-183 according to the g* power software as follows,

F tests - ANOVA: Fixed effects, special, main effects and interactions

Analysis:     A priori: Compute required sample size 

Input:                         Effect size f                         =      0.25

                                   α err prob                            =      0.05

                                   Power (1-β err prob)           =            0.8

                                   Numerator df                      =            2

                                   Number of groups               =            3

Output:                      Noncentrality parameter λ  =      9.8750000

                                   Critical F                             =      3.0543850

                                   Denominator df                   =            155

                                   Total sample size               =            158

                                   Actual power                      =      0.8021998

According to the result of G*power program, the calculated total sample was 158 so we design to enroll as approximately 180 participants (n=60/group) with cover 10% drop-off.

2.Studied population, the population sample recruitment has been performed from, shall be described. e.g. patients referred to a specific hospital for a specific perpous, volunteer health care providers, workers in an institution.
   Response 2: We added a few sentences in line no. 87-90.

Reviewer 2 Report

The authors answered satisfactorily all the questions raised by this reviewer. The manuscript was improved and is now suitable for publication.

Author Response

We appreciate on your comments

Round 3

Reviewer 1 Report

Decision: Accept